# Nematophagous Fungi: A Review of Their Phosphorus Solubilization Potential

**DOI:** 10.3390/microorganisms11010137

**Published:** 2023-01-05

**Authors:** Marcos Vera-Morales, Segundo E. López Medina, Jaime Naranjo-Morán, Adela Quevedo, María F. Ratti

**Affiliations:** 1Escuela de Postgrado, Universidad Nacional de Trujillo, Jr. San Martin 392, Trujillo 13007, Perú; 2Escuela Superior Politécnica del Litoral, ESPOL, Centro de Investigaciones Biotecnológicas del Ecuador, CIBE, Campus Gustavo Galindo Km. 30.5 vía Perimetral, Guayaquil EC090112, Ecuador; 3Laboratorio de Biotecnología Vegetal, Ingeniería en Biotecnología, Facultad Ciencias de la Vida, Campus María Auxiliadora, Universidad Politécnica Salesiana (UPS), Km 19.5 Vía a la Costa, Guayaquil P.O. Box 09-01-2074, Ecuador; 4Escuela Superior Politécnica del Litoral, ESPOL, Facultad de Ciencias de la Vida, FCV, Campus Gustavo Galindo Km. 30.5 vía Perimetral, Guayaquil EC090608, Ecuador

**Keywords:** biocontrol, nematophagous fungi, phosphorus, solubilization

## Abstract

Nematophagous fungi (NF) are a group of diverse fungal genera that benefit plants. The aim of this review is to increase comprehension about the importance of nematophagous fungi and their role in phosphorus solubilization to favor its uptake in agricultural ecosystems. They use different mechanisms, such as acidification in the medium, organic acids production, and the secretion of enzymes and metabolites that promote the bioavailability of phosphorus for plants. This study summarizes the processes of solubilization, in addition to the mechanisms of action and use of NF on crops, evidencing the need to include innovative alternatives for the implementation of microbial resources in management plans. In addition, it provides information to help understand the effect of NF to make phosphorus available for plants, showing how these biological means promote phosphorus uptake, thus improving productivity and yield.

## 1. Introduction

Pathogenic threats and nutritional deficiency are the main problems that affect yield and productivity in plant crops [1], increasing the need to apply new microbial agents to solve both obstacles [2]. Thus, NF have been investigated because of their dual ability as nematode controllers and as plant growth promoters that increase the bioavailabilty of nutrients [3]. 

Phosphorus (P) is the second most important element for plants after nitrogen. It regulates metabolism and plant development and growth, and an adequate supply is needed to keep these metabolic functions [4]. In addition, it is found in abundance in soil, although its uptake is low, which results in a limitation for the productivity of agricultural crops [5].

P can be solubilized by several soil microorganisms (saprotrophic bacteria and fungi) [6]. Phosphate-solubilizing microorganisms (PSM) play an important role in soil by mineralizing organic P, solubilizing inorganic P minerals, and storing P in biomass [7], thus improving plant growth and yield [8]. PSM can solubilize phosphorus by secreting protons and producing organic anions, such as citrate, oxalate, and gluconate [9]. However, the amount of P solubilization depends on the microbial strain and the relationship between carbon sources, P, and organic acid production [10].

Bacteria can release organic acids that solubilize P or produce acid and alkaline phosphatases that mineralize organic P [11]. There are bacterial genera that are known to be very efficient in these mechanisms, such as *Arthrobacter*, *Bacillus*, *Burkholderia*, *Natrinema*, *Pseudomonas*, *Rhizobium*, and *Serratia* [12]. Bacteria are superior to fungi in colonizing plant roots, but are less tolerant to acids [13], so fungi have great potential to solubilize P in acidic conditions [14]. Fungi, in fact, have a higher capacity to release P compared with bacteria [15]. They use different mechanisms such as the secretion of fatty acids, the production of enzymes, and the discharge of metabolic substances (siderophores) that can rapidly metabolize P [16].

Despite the numerous studies on NF as biocontrol agents, the comprehensive and complete understanding of the mechanisms of action is still incipient [17]. In general, the interactions of NF have been evaluated to reduce the densities of plant pathogenic nematodes [18]. Yet, they present other benefits such as the synthesis of phytohormones, as well as favoring the absorption of P [19]. The inoculation of P solubilizing microbial agents in seeds, crops, and soil aims at improving agricultural production without affecting soil health [20]. Bioformulations based on fungal strains that increase plant yield and are also effective at solubilizing nutritional elements are needed [21]. Therefore, the objective of this review is to broaden the understanding of NF and their potential use in P solubilization, focusing on their mechanisms of solubilization and the benefits provided in the nutrition of agricultural crops.

## 2. Mechanisms of P solubilization by NF

Plants can harness various forms of P; nevertheless, roots take up negatively charged forms of orthophosphate ions (HPO_4_^2−^- and HPO_4_^−^) depending on the soil pH [22]. Although agriculture practices introduce considerable amounts of phosphate in soil, only 15 to 30% of P is taken up by plants [23]. It generally binds to iron and aluminum oxides and hydroxides, or to calcium in calcareous soils, becoming unavailable for roots due to chemical precipitation or physical adsorption [24]. Therefore, the application of P itself does not contribute to the improvement of agricultural production systems (Figure 1).

Several microorganisms have been studied in relation to the solubilization of P. Fungi and bacteria use different mechanisms on soluble phosphates, especially acidification of the medium, chelation, exchange reactions, and the production of solubilizing enzymes and of various organic acids [4,8,20,25]. Therefore, some species are able to solubilize and mobilize P for plants, including, several nematode controlling fungi. An example is *Trichoderma harzianum*, which can acidify the medium, produce chelating metabolites, and produce a redox activity (capable of reducing Fe and Cu) [26].

### 2.1. Organic Acids

Organic acids or anions range from fatty acids to secondary metabolites [22]. They are by-products derived from bacteria and fungi that are able to mobilize P in their microenvironment. Organic acids improve P release so that it can be absorbed by plants [27]. Filamentous fungi use several mechanisms to solubilize minerals and they produce high amounts of organic acids [28]. Species of the genera *Aspergillus*, *Penicillium*, *Trichoderma*, and *Fusarium* solubilize P by producing different types of organic acids [14,29,30,31,32]. Furthermore, it is possible that some filamentous fungi such as *Aspergillus niger* solubilize P when exposed to high salt contents (4% NaCl) [33]. Even on ferric phosphate substrates, *A. niger* can secrete large amounts of citric and oxalic acids to release P [34]. Therefore, the potential of filamentous fungi in the agrobiotechnology industry is increased, especially because of their properties, for the production of organic acids [35].

### 2.2. Solubilizing Enzymes

Organic P from animal and plant residues can be mineralized by the action of phosphatases [36], such as phytases that hydrolyze the phospho-monoester bonds present in phytates [37]. Nematode controlling fungi can produce phytases during their development or the production of trapping structures [38]. The nematode-capturing fungus *Arthrobotrys oligospora* forms phytases while building adhesive networks, significantly increasing the release of P [39]. Among the genera of NF that mainly produce phytases and phosphatases under laboratory conditions are the filamentous fungi *Aspergillus*, *Penicillium*, and *Trichoderma* [40].

### 2.3. Siderophore Production

These are low molecular weight organic iron chelating compounds. They can be produced by bacteria, fungi, and plants [40]. Among the most studied fungi for siderophore production are *Aspergillus fumigatus* and *Aspergillus nidulans*, which have 55 similar types of siderophores [41,42]. In addition, *Penicillium* produces siderophores, capable of solubilizing tricalcium phosphate, even in contaminated soils, opening new opportunities in the field of phytoremediation and agrobiotechnology [43]. Therefore, the use of siderophores is increasing in use as a new alternative for agriculture, to replace pesticides and synthetic fertilizers [44].

## 3. Potential of NF in P Solubilization 

NF can be used in agriculture through the massive production of infective spores or through formulations that improve growth, viability, and efficacy [45]. They use several mechanisms of recognition, signaling, differentiation, and penetration of the cuticle or egg shell of nematodes through mechanical and enzymatic actions [46]. For the purposes of this study, NF are classified according to their mechanism of attack, including trapping, endoparasitic, egg parasitic, toxin-producing, and special device-producing species [47]. NF interact with the environment, performing essential functions to maintain the stability of food webs and the cycling of nutrients in soil [48].

NF widely used in management plans as biological control agents include *Purpureocillium lilacinum*, *Pochonia chlamydosporia*, *Trichoderma harzianum*, *Arthrobotrys* spp., *Hirsutella* spp., etc. [49]. Some of these fungi are widely used in agriculture due to their ability to infect and kill arthropods [50]. They are recognized as biopesticides in the management of destructive pests. Some notable examples include *Beauveria bassiana* [49], *Aspergillus fijiensis* [51], *Cladosporium tenuissimum*, and *Penicillium citrinum* [52]. Among NF, some have been reported as P solubilizers. *Trichoderma* is one of the most studied genera and is considered as a growth promoter, pathogen controller, and nutritional promoter of plants [53,54]. Furthermore, some NF can solubilize P, even under low temperature conditions [55].

## 4. Trapping Nematodes—Predatory Fungi 

Fungi that capture nematodes have a double function: controlling the populations of infective nematodes and also acting as biofertilizers when applied in soil, being able to solubilizing and mobilize some important nutrients [56,57]. The adhesive networks building fungus *Duddingtonia flagrans* have been shown to solubilize P both in laboratory [56] and in greenhouse conditions [2]. Likewise, *A. oligospora* has been used successfully in the solubilization of phosphate rock in in vitro and in vivo conditions (Figure 2), showing promising results in the uptake of P by plants [58].

## 5. Endoparasitic and Egg-parasitic Fungi 

Endoparasitic fungi are obligate pathogens that infect nematodes. Conidia of the obligate endoparasitic fungus *Drechmeria coniospora* are ingested or adhere to the cuticle of its hosts [59]. The spores produce a germination tube that enters the nematode, producing a mycelium that, by degrading internal tissues, exploits the host nutrients for growth [60]. One of the most studied fungal parasites of nematode eggs is the fungus *Pochonia chlamydosporia*, able to reduce populations of several plant parasitic nematodes, including species of *Globodera*, *Heterodera*, *Meloidogyne*, *Nacobbus*, and *Rotylenchulus* [61]. *Pochonia chlamydosporia* benefits the plant, as it is able to control nematode infections, shortening the flowering and fruiting times (up to 5 and 12 days), as well as significantly increasing root growth [62]. This parasitic fungus produces phosphatases, organic acids, and propionic acid, which promote the depolymerization of phosphate compounds [63,64]. Furthermore, in association with the predatory fungus *D. flagrans*, they can significantly increase nutrient uptake, especially P, by up to 70% in tomato plants [3]. Therefore, research studies suggest the use of bioformulated *P. chlamydosporia* for the management of phytoparasitic nematodes, with the potential to support plant nutrition [65]. 

*Purpureocillium* is a further fungal genus used for the reduction of phytoparasitic nematode eggs and females. Some species attack the gall-forming nematodes *Meloidogyne* spp. [66], and show a P solubilization capability in laboratory conditions [67], as well as promoting tomato seedling growth in greenhouse conditions [68]. These fungi may secrete hydrolytic enzymes and siderophores that solubilize P and promote plant growth [69].

## 6. Toxin Production

Toxin-producing fungi are characterized by the release of compounds that paralyze nematodes and produce a rapid and systemic cell necrosis in multiple tissues throughout their bodies. In the order of the Agaricales, one representative is *Pleurotus ostreatus*, which is a toxin-producing fungus capable of producing necrosis after the contact of hyphae with the cilia of the sensory system of nematodes [70]. The fungus has P solubilizing properties [71] and is applied as a biofertilizer based on biomasses obtained as mycelial by-products [72]. 

*Piriformospora indica* is an endosymbiotic fungus belonging to the order Sebacinales (Basidiomycota), capable of producing metabolites that inhibit nematode populations [73]. This species has shown synergy with phosphate-solubilizing bacteria, sustaining the growth of chickpea plants [74]. Studies have indicated that the transfer of P by *P. indica* to the host plant has a significant effect on its development [75].

## 7. Filamentous Fungi with Nematophagous Activity 

In addition to the examples mentioned above, there are a variety of fungi that do not use mechanisms based on capture, parasitism, toxins, or special devices, so it is presumed that hydrolytic enzymes are a key in the nematode infection process [76]. Among the known genera that work as both biocontrol and P solubilizers are *Trichoderma*, *Fusarium*, *Penicillium*, and *Aspergillus.* Filamentous fungi can secrete hydrolytic enzymes, organic acids, and low molecular weight natural products, which confer several functions, including P solubilization [77], and are able to solubilize calcium and iron phosphates [78]. Thus, they hold great potential for the development of biofertilizers, contributing to soil fertility and promoting plant growth [79], which are essential in sustainable agriculture (Table 1) [80]. 

Filamentous fungi such as *Trichoderma* spp. are often dominant in soil microbial communities and have the ability to colonize roots [81]. *Trichoderma* spp. are marketed all over the world for their potential as biocontrol and biostimulant agents for numerous agricultural crops. The application of *Trichoderma* spp. and related metabolites allegedly improves crop productivity, nutrient supply, and defense against plant pathogens [54,82], as it combines P solubilization and nematode predation.

**Table 1 microorganisms-11-00137-t001:** Fungi known as nematophagous and P solubilizers.

Type of Fungi	Fungi	Structure Involved	Substance That Is Solubilized	Solubilization Mechanism	Reference
Predatory	*Arthrobotrys oligospora*	Adhesive networks	Phosphate rock	pH of the culture medium.	[58]
	*Arthrobotrys conoides* and *Duddingtonia flagrans*	Adhesive networks Adhesive networks	Tricalcium, zinc, and aluminum phosphate phosphate rock	pH of the culture medium. Production of organic acids	[56]
	*Duddingtonia flagrans*	Adhesive networks	Phosphorus	-	[2,3]
Nematode egg-parasitic	*Pochonia chlamydosporia*	Appressoria and hyphae	Phosphorus	Phosphatases and organic acids	[3,62,63]
	*Purpureocillium variotii*	Toxic metabolites	Phosphorus	Siderophores and pH of the culture medium	[68]
	*Purpureocillium hepiali*	Toxic metabolites	Phosphate	pH of the culture medium	[67]
	*Purpureocillium lilacinum*	Toxic metabolites	Calcium and iron phosphate	pH of the culture medium and organic acids	[83]
Toxin producing	*Piriformospora indica*	Metabolites	Organic and inorganic phosphorus	pH of the culture medium	[73,75,84]
	*Pleurotus ostreatus*	Toxins	Phosphate rock	Organic acids: tartaric, malic, citric, lactic, succinic and four unknown acids	[71,85]
Filamentous fungi with indetermined nematophagous mechanism	*Trichoderma harzianum*	Hydrolytic enzymes	Calcium phosphate	pH of the culture medium, production of chelating metabolites, and redox activity	[26]
	*Trichoderma asperellum*	Hydrolytic enzymes	Monopotassium phosphate Phytate Phosphate rock	pH of the culture medium and organic anions	[86]
	*Trichoderma* spp.	Hydrolytic enzymes	Phosphate	Organic acids	[87]
	*Aspergillus niger*, *Penicillium canescens*, *Eupenicillium ludwigii*, and *Penicillium islandicum*	Hydrolytic enzymes	Phosphate rock	pH of the culture medium, acids: oxalic, citric, and gluconic	[31]
	*Aspergillus*, *Penicillium, Trichoderma*, *Fusarium*, *Mucor*, *Ovularopsis*, *Tritirachium*, and *Geotrichum*	Hydrolytic enzymes	Phosphate rock Tricalcium phosphate	Acids: fumaric, acetic, gluconic, lactic, and succinic	[29]
	*Penicillium expansum*, *Mucor ramosissimus*, and *Candida krissii*	Hydrolytic enzymes	Phosphate rock	Acids: citric, oxalic, and gluconic	[79]
	*Penicillium guanacastense*	Hydrolytic enzymes	Organic and inorganic phosphorus	Acids: gluconic, oxalic, lactic, and malonic	[32]
	*Penicillium bilaji* and *Penicillium cf.fuscum*	Hydrolytic enzymes	Phosphate rock	Organic acids	[30]
	*Aspergillus niger*	Hydrolytic enzymes	Various forms of soluble P calcium iron aluminum phosphate	Acids: gluconic, oxalic, tartaric	[14,33]
	*Aspergillus aculeatus*	Hydrolytic enzymes	Phosphate rock	pH of the culture medium	[88]
	*Rhizopus stolonifer*, *Aspergillus niger*, and *Alternaria alternata*	Hydrolytic enzymes	Calcium phosphate	Organic acids	[89]
	*Aspergillus tubingensis*	Hydrolytic enzymes	Phosphate rock	pH of the culture medium	[90]
	*Penicillium oxalicum* and *Aspergillus niger*	Hydrolytic enzymes	Tricalcium phosphate	Organic acids	[91]
	*Penicillium bilaii*	Hydrolytic enzymes	Calcium phosphate	Acids: citric and oxalic	[92]
	*Aspergillus niger* and *Penicillium italicum*	Hydrolytic enzymes	Tricalcium phosphate	pH of the culture medium	[93]
	*Penicillium oxalicum*, *Trichoderma virens*, and *Aspergillus*	Hydrolytic enzymes	Insoluble tricalcium phosphate, soluble dipotassium hydrogen phosphate	pH of the culture medium and organic acids	[94]
	*Penicillium*	Hydrolytic enzymes	Phosphate	pH of the culture medium	[95]
	*Penicillium oxalicum*	Hydrolytic enzymes	Phosphate	Organic acids	[96]
	*Paecilomyces*, *Trichoderma*, *Aspergillus*, *Fusarium*, and *Gongronella*	Hydrolytic enzymes	Calcium phosphate Iron phosphate		[78]

## 8. Agricultural Solutions: Application of NF in P Solubilization

Although P is added to the soil of agricultural crops, chemical fertilizer synthesis is an energy-consuming process with long-term impacts on the environment in terms of eutrophication, fertility decrease, and carbon footprint [97]. Nematophagous species have been used for plant growth promotion and to sustain yields, especially members of the genera *Trichoderma*, *Purpureocillium*, *Pochonia*, *Fusarium*, *Arthrobotrys*, and *Verticillium* [98]. 

Several NF are beneficial because of the production of phytohormones, antibiotics, or siderophores, which benefit long-cycle and short-cycle crops [99] (Table 2), as shown by the P solubilization activity in alkaline soils, which increase the yields of corn and wheat [100,101]. These NF potentially shorten the maturity period of crops, improve fruit quality, increase the availability of soluble P, and improve soil biodiversity [102]. Thus, there is a positive link between the application of these fungal strains and the P content in soil and plants [103].

Currently, several investigations suggest the use of fungi as bioinoculants to increase the yield in agricultural crops [104], as well as ornamentals. P solubilization has been in fact also reported in shrubs within ornamental greenhouses, as demonstrated by the application of *Mortierella* sp. on seedlings of *Leucaena leucocephala* [105].

**Table 2 microorganisms-11-00137-t002:** List of P-solubilizing NF and related benefits in agricultural crops.

P-Solubilizing Species	Crop	Plant Benefits	Reference
*Geomyces pannorum*, *Paecilomyces carneus*	*Avena sativa*	Increase availability of phosphorus in the soil and mitigate phytoparasitic nematodes	[106]
*Duddingtonia flagrans* *Pochonia chlamydosporia*	Soy bean and tomato	Reductions in the number of eggs and galls per gram of root and increased nutrient content in roots	[2,3]
*Pochonia chlamydosporia*	Tomato	Increase in secondary roots and increase in the total weight root of seedlings Reduction in flowering and fruiting times and greater weight of mature fruits	[62]
*P. chlamydosporia*	Maize Cowpea	Promoting root growth	[107]
*Purpureocillium lilacinum* *Purpureocillium lavendulum* *Metarhizium marquandii*	Maize Beans Soy bean	Plant growth promotion and availability of P and N	[104]
*Purpureocillium lilacinum*	Tomato	Plant growth promotion and phosphorus solubilization	[69]
*Pleurotus ostreatus*	*Maize*	Increase in root and shoot lengths, fresh and dry root weights, fresh and dry shoot weights, chlorophyll content, and nutrient uptake	[71]
*Trichoderma*, *Purpureocillium*	Banana	Active production of indole-3-acetic acid IAA and solubilize insoluble phosphate	[99]
*Trichoderma*	Soy bean	Plant growth promotion increase of P uptake efficiency	[82,87]
*Aspergillus niger*, *A. fumigatus*, *Penicillium pinophilum*	Wheat and faba bean	Yield of wheat grains and faba bean seed production	[108]
*Penicillium oxalicum*	Wheat and maize	Replace chemical fertilizer in alkaline soils. Improved crop production	[101]
*Penicillium* sp. *Penicillium oxalicum*	Maize	Increased uptake of P by plants and increased availability of P in soil	[100]
*Penicillium* sp. *Aspergillus foetidus*	*Sorghum bicolor* L.	P uptake and increased growth	[109]
*Penicillium expansum*, *Mucor ramosissimus*, *Candida krissii*	Wheat	Plant growth promotion, P available in soil, and P absorption	[79]
*Penicillium oxalicum*	Rapeseed	Solubilize inorganic P and mineralize organic P	[110]
*Mortierella capitata*	Maize	Increased biomass, chlorophyll, and gibberellic acid	[111]
*Mortierella* sp.	Avocado	Plant growth promotion and P uptake	[112]

## 9. Future Research Perspectives 

The main P source in the world, phosphate rock, is fundamental for food production. It is strongly threatened as it is a finite resource. P, for the most part, is limited to a few countries in the world, particularly Morocco, which holds 75% of the world reserve [113]. Therefore, in recent years, the challenge of ensuring sustainable global P management to achieve world food security has been evidenced [114]. Given the current agricultural practices dependent on the continuous supply of commercial P-based fertilizers [115], the need to search for new sustainable strategies to manage the P availability in agricultural fields has increased [116]. Therefore, biological activators must be used to accelerate the bioavailability of P for plants, making NF important not only for the biological control of nematodes, but also for P solubilization. Their applications are also in accordance with sustainable agricultural practices for developing countries [117].

Currently, microbial-based fertilizers are not only considered for productivity and economic benefits, but also for their use as environmental-friendly products [18]. The aim is to guarantee less damage to water quality, increase nutrients recycling, reduce the consumption of resources, improve soil health, and increase the biodiversity of beneficial microorganisms [118]. Therefore, there is a demand to join efforts towards the search and discovery of microfungi, especially from little-explored natural regions [119], with the potential to sustain crop nutrition [120]. In fact, the excessive use of fertilizers affects soil health, adding to the presence of pathogens and pests [121]. 

Filamentous fungi with a dual activity (nematicide and P solubilizers) have versatile capacities to synthesize biocompounds such as enzymes, organic acids, and metabolites [122]. Especially, for P bioavailability, they can biologically produce high concentrations of organic acids, offering new knowledge through the detection of genes related to the mechanisms of phosphate solubilization [24,120,123]. Biotechnology applications may offer sustainable solutions based on microorganisms that are tolerant to new environmental conditions, including the microbiota that favors plant nutrition [124].

## 10. Conclusions

Several NF have the ability to suppress nematode parasites and improve the uptake of nutritional elements in order to promote plant growth and development. They have been successfully applied as biofertilizers and biocontrol agents to establish beneficial ecological relationships within their environment. For P solubilization, they rely on different mechanisms, such as a decrease in pH and the production of siderophors, organic acids, and enzymes. The wide variety of mechanisms for P solubilization in NF can be harnessed to reduce the dependence on P-based fertilizers in agriculture. The availability of new enzymes such as phytases promotes the search for beneficial microorganisms to develop environmentally-friendly and sustainable management plans.

## Figures and Tables

**Figure 1 microorganisms-11-00137-f001:**
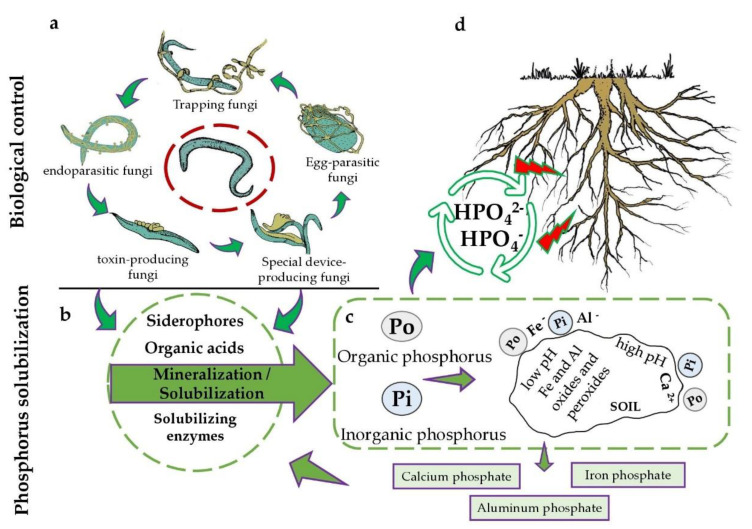
Systematic diagram of P solubilization by fungi used in the biological control of nematodes. (**a**) Nematodes can be parasitized and predated by (in clockwise order) trapping fungi, egg-parasite fungi, special device-producing, or toxin-producing and endoparasitic species. (**b**) Compounds produced by fungi. (**c**) Chemical reactions of P in soil, binding with elements such as calcium, iron, and aluminum that can then be mineralized or solubilized by the NF compounds. (**d**) Roots absorbing negatively charged orthophosphate ions.

**Figure 2 microorganisms-11-00137-f002:**
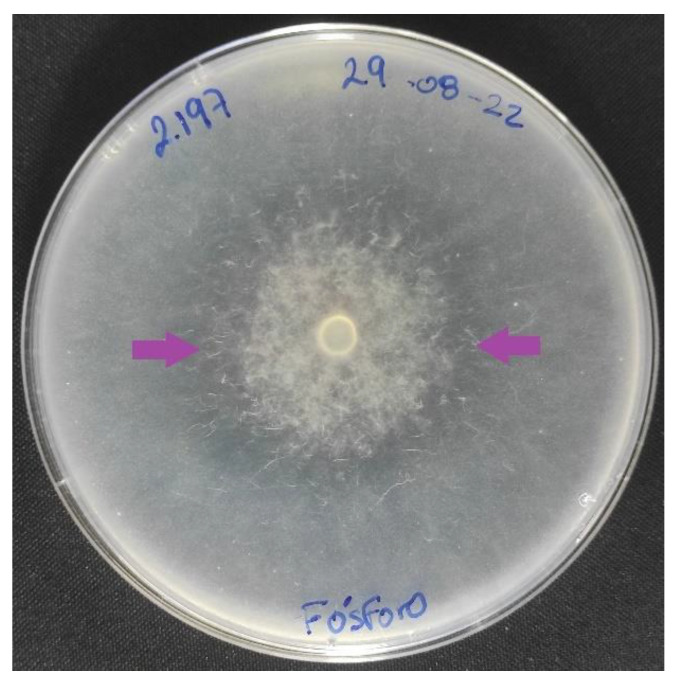
Nematode-capturing fungus *Arthrobotrys oligospora*. Phosphate solubilization zone on Pikovskaya agar plates, as shown by the halo (arrows).

## Data Availability

Not applicable.

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
