# Peer review of "Nematophagous Fungi: A Review of Their Phosphorus Solubilization Potential"

_microorganisms, 2023, doi:10.3390/microorganisms11010137_

Round 1

Reviewer 1 Report

The manuscript reviews the role of nematophagous fungi in recycling and mobilizing phosphorus in soil and rhizosphere. The paper does not cover the entire literature data about this topic, for example some studies focusing on such aspects were not considered (i.e. Ann. Appl. Biol., 164, 232-243). However, it may result of interest for a wide audience of researchers interested in soil microbial ecology and plant nutrition. The paper may be improved by simplifying the terminology (i.e. use NF instead of repeating "nematophagous fungi"). There are some minor issues that should be resolved before publication, mostly dealing with style. Figure 2 is not suitable for publication as it shows nematodes trapped and conidia, however such images are already widely present in literature and internet. It is suggested to use only Fig. 2a and eliminate 2b and 2c (moreover these two images do not match the legend text, which is inverted). In the reference list keep the species names with minor initials. In text, use P or phosphorus, not both terms (the first one is preferable). Nematodes are parasites, not pathogens, the latter term is used for  microorganisms, so avoid using "disease/pathogen" and use "parasitism/parasite". Pay attention to the species latin names (i.e. Arthrobotrys oligospora, not oligosporus). Other amendments are shown on the attached, revised file. The paper may be accepted after the changes requested are applied.

Author Response

The authors wish to express their gratitude for the suggested recommendations to improve the manuscript. Thus, we have added a section related to the recommended bibliography (Ann. Appl. Biol., 164, 232-243). Furthermore, terminologies have been simplified and style-related issues have been resolved. In Figure 2, as suggested, we left only the panel recommended by the reviewer, that is, Fig. 2a. In the list of references, scientific names have been kept in lower case. The terms “parasitism/parasite” have been used to name nematodes. The Latin names of the species and the recommendations in the attached file were modified.

Reviewer 2 Report

Aprovado

Author Response

The authors wish to thank you for the recommendations, and we send the new version of the manuscript.

Reviewer 3 Report

I recommend the publication of the article because the scientific experimentation related to studies on nematophagous fungi activity and fungal-plant interaction is really interesting and topical. In particular, their activity in phosphorous solubilisation, organic acid production, enzyme production and their activity in the soil.

The aim and objectives of the article have been stated and are very fascinating.

The use of nematophagous stimulating fungi for plant growth and defence is an important topic especially in organic farming with regard to the reduction of synthetic products and chemical fertilisers. Furthermore, many aspects of their activity have not yet been highlighted. The work is certainly of international interest and the format applied is certainly suitable for a research paper. The work is original, of particular interest and can certainly stimulate research on this topic. The length of the article is good for the journal and the graphs and tables are clear and easy to understand. The conclusion summarises the aims of the work and future prospects.

Author Response

The authors wish to express their gratitude and we send you the new version of the manuscript.

Reviewer 4 Report

The review article is clear, relevant for the field and presented in a well-structured manner.

I have some suggestions to improve the quality of manuscript before publication.

Title:  Written well.

Keywords don’t use keywords which already mentioned in title.

Introduction is in general correct however, comprehensive review of modern literature required that will clearly and convincingly demonstrate the significance in the management of arthropod pests.

Please read some recent studies about Aspergillus and Penicillium and other fungi as an entomopathogenic fungi and add in the table

First record of Aspergillus fijiensis as an entomopathogenic fungus against asian citrus psyllid, Diaphorina citri Kuwayama (Hemiptera: Liviidae). Journal of Fungi, 2022.

Effectiveness of Entomopathogenic Fungi on Immature Stages and Feeding Performance of Fall Armyworm, Spodoptera frugiperda (Lepidoptera: Noctuidae) Larvae. Insects, 2021.

Bioassays of Beauveria bassiana Isolates against the Fall Armyworm, Spodoptera frugiperda Journal of Fungi, 2022.

References: Most of the cited references are quite current. There is a normal, justified number of self-citations. Please check the scientific name as in some reference the scientific name needs to be italicize.

General: Revise scientific names: in several lines of the text scientific names are written in regular letters and not in italics.

English Language: Even English language problems are going to be addressed at a later stage by the journal internal staff, it is a point that must be emphasized. There are some punctuation and grammar mistakes.

Author Response

The authors wish to express their gratitude for the suggested recommendations. We have added a paragraph from lines 128-132 where the recommended references were included. The references have been updated especially in the spelling mistakes in scientific names. Finally, modifications have been made to improve English punctuation and grammar.
